# PetBuddy

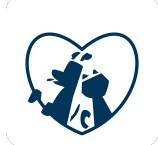

## Aplikacja webowa do ułatwienia komunikacji między właścicielami zwierząt domowych, a potencjalnymi opiekunami



**Autorzy**: Łukasz Cyndzer● · Jakub Gawron● · Karol Piłat● · Jakub Staniszewski●
**Opiekun:** Ireneusz Jóźwiak

**Streszczenie**

Projekt ma na celu usprawnić komunikację między właścicielami zwierząt domowych a potencjalnymi opiekunami. PetBuddy umożliwia organizację opieki nad zwierzęciem w czasie nieobecności jego właściciela, niezależnie od tego, czy wynika to z obowiązków zawodowych, czy z dłuższej podróży wakacyjnej. Jednocześnie aplikacja daje możliwość zarobku osobom, które posiadają chęci i możliwości opieki nad cudzym zwierzęciem, poszerzając w ten sposób dostępność i zasięg rynku usług związanych z opieką nad zwierzętami. Wspomniane cechy naszej aplikacji sprzyjają radzeniu sobie z wyzwaniami dzisiejszych czasów w pogodzeniu pracy i podróży z posiadaniem zwierząt.

## 1. Cel

W Polsce zwierzęta domowe odgrywają znaczącą rolę w życiu wielu gospodarstw domowych. W 2024 roku około 49% polskich domów posiada co najmniej jednego psa [1], a łączna liczba zwierząt domowych wynosi około 18,336,600 [1]. Tak duża populacja zwierząt domowych wiąże się z koniecznością zapewnienia im odpowiedniej opieki, która obejmuje zarówno podstawowe potrzeby, jak i opiekę w sytuacjach, gdy właściciele nie mogą się nimi zajmować.

Jednocześnie Polska jest jednym z krajów Unii Europejskiej o najwyższym średnim tygodniowym czasie pracy. Zgodnie z danymi Eurostatu z 2023 roku przeciętny pracownik w Polsce spędza w pracy 39,3 godziny tygodniowo [2], co plasuje Polskę na szóstym miejscu w Unii Europejskiej pod tym względem. Częste podróże związane z pracą, wakacjami lub innymi obowiązkami dodatkowo pogłębiają wyzwanie związane z organizacją opieki nad zwierzętami.

W odpowiedzi na te potrzeby chcieliśmy stworzyć aplikację webową, która ułatwi właścicielowi, zwanego dalej także klientem, zorganizowanie opieki dla swojego zwierzęcia. System ma na celu zapewnienie wygodnego i intuicyjnego narzędzia do wyszukiwania odpowiednich opiekunów według zadanych filtrów, m. in. po miejscowości, dacie czy cechach zwierzęcia. W ramach wyszukiwania, możliwe jest także sprawdzenie przybliżonej lokalizacji opiekunów na mapie. Ponadto, aplikacja usprawnia samą organizację opieki poprzez prosty formularz zgłoszeniowy, jak i daje możliwość sprawnej negocjacji ceny opieki. System zapewnia również efektywną komunikację między opiekunem, a właścicielem zwierzęcia, realizowaną przez czat, który jest dostępny nawet przed złożeniem rezerwacji opieki. Kolejne cele skupiają się wokół samego opiekuna. Jednym z nich jest umożliwienie zarobienia dodatkowych pieniędzy przez świadczenie płatnej usługi opieki nad zwierzętami. Co więcej, aplikacja ułatwia ogłaszenie i oferowanie swoich usług.

## 2. Prace związane z tematem

### 2.1. Analiza istniejących rozwiązań

#### 2.1.1 Petsy [3]

Jest to aplikacja pozwalająca na znalezienie opiekuna dla zwierzęcia domowego. Względem naszego projektu wspomniana aplikacja różni się tym, że ogranicza się jedynie do dwóch rodzjów zwierząt: psa i kota. Co więcej, nie daje możliwości logowania przez Google, posiada wysokie prowizje, skupia się na większych miejscowosciach oraz nie umożliwia negocjacji cen między klientem, a opiekunem. Dużą zaletą Petsy jest zapewnienie pomocy weterynaryjnej, wsparcia behawiorysty i ubezpieczenie.

### 2.1.2   Animalhotels [4]

Kolejnym rozwiązaniem jest animalhotels, które skupia się nie tylko na opiekunach, ale również na hotelach dla zwierząt. Aplikacja pozwala na przeglądanie profili opiekunów, gdzie można znaleźć m.in. informacje na temat oferowanych usług czy pokazanym na mapie przybliżonym miejscu zamieszkania. Pozwala również na komunikację między klientem, a opiekunem lub hotelem oraz dokonywanie płatności wewnątrz aplikacji. Szczególnie przydatną funkcją, którą dostarcza animalhotels jest formularz, dotyczący rodzaju zwierzęcia oraz miejsca i terminu opieki. Po jego wypełnieniu aplikacja wysyła zapytanie do opiekunów mieszkających w pobliżu, czy mają możliwość zaopiekowania się danym zwierzęciem. Podobnie jednak jak w Petsy, animalhotels nie daje możliwości logowania przez Google, ma wysokie prowizje, skupia się na wiekszych miejscowosciach i nie daje możliwości negocjacji cen między klientem, a opiekunem.

### 2.1.3   Rover [5]

Trzecią aplikacją, która umożliwia odnalezienie opieki dla zwierzęcia domowego jest Rover. Podobnie jak Petsy, zawiera oferty dotyczące jedynie kotów i psów. Posiada jednak wiele rodzajów usług: wyprowadzanie psów, opieka nad psem w domu opiekuna lub w domu klienta. Co więcej, daje możliwość znalezienia cotygodniowej opieki, obsługuje płatności wewnątrz aplikacji, pozwala na logowanie za pomocą emaila, Google, Facebooka i Apple oraz dostarcza możliwość zapisania się na kurs szkolenia psa. Dużą zaletą wspomnianej aplikacji jest zapewnienie ubezpieczenia dla klienta i jego zwierzęcia na opiekę weterynaryjną podczas rezerwacji.

## 2.2.   Wybór technologii

### 2.2.1   Backend

Jeśli chodzi o wybór technologii używanej przez nas w aplikacji backendowej, to zdecydowaliśmy się na framework Spring [6] oparty na języku Java. Wybór ten znacznie ułatwia implementację systemu jeśli chodzi o dostęp do zasobów bazy danych czy też walidację i szybką obsługę zapytań w czasie rzeczywistym. W szczególności korzystamy z Hibernate, popularnej biblioteki ORM (Object-Relational Mapping), wspieranej przez framework, która umożliwia szybkie tworzenie i prototypowanie bazy danych. Dzięki temu proces zapisu i odczytu danych jest prostszy i bardziej efektywny, eliminując konieczność pisania skomplikowanych zapytań SQL, co znacząco zwiększa wydajność i ułatwia utrzymanie kodu.

Dodatkowo, Spring zapewnia wsparcie dla WebSocketów, co umożliwia tworzenie aplikacji opartej na komunikacji w czasie rzeczywistym. Dzięki temu możemy zaimplementować funkcje takie jak czaty, powiadomienia czy dynamiczne aktualizacje danych, co zapewnia użytkownikom interaktywne i responsywne doświadczenie. Wsparcie dla WebSocketów w Springu pozwala na łatwą integrację tej technologii z resztą aplikacji, umożliwiając szybkie przesyłanie danych między klientem a serwerem. Technologię tą używamy w implementacji czatu w czasie rzeczywistym oraz wysyłania powiadomień użytkownikowi zapewniając szybką i natychmiastową wymianę danych między klientem a serwerem.

### 2.2.2   Keycloak

Następną technologią, którą użyliśmy jest Keycloak - narzędzie służące do zarządznaia użytkownikami, wraz z ich uwierzytelnianianiem oraz autoryzacją. Wybraliśmy Keycloak'a ze względu na jego szeroki wachlarz funkcji i możliwość konfiguracji. Jest to również technologia zaufana, ciągle rozwijana i otwarto-źródłowa [7]. Zależało nam aby dać użytkownikowi możliwość logowania się za pomocą nie tylko e-maila i hasła, ale również za pośrednictwem portali społecznościowych, takich jak Google, co Keycloak realizuje poprzez implementację protokołu OAuth2.

### 2.2.3   PostgreSQL

Jeśli chodzi o bazę danych, to technologią którą wybraliśmy w celu jej zrealizowania jest otwartoźró-dłowe oprogramowanie [8] - PostgreSQL. Ze względu na przewidywany tabelaryczny charakter danych które przechowujemy, uznaliśmy że baza danych relacyjna doskonale się sprawdzi w naszym projekcie. Dodatkowo, PostgreSQL jest to jeden z liderów na rynku jeśli chodzi o bazy danych tego typu oraz jest wspierany przez Keycloak'a podczas gdy wsparcie dla MySQL nie jest prewidziane w kontekście długoterminowym.

### 2.2.4   Firebase

Technologią do zapisu zdjęć użytkowników, na którą się zdecydowaliśmy został Firebase, a uściślając - usługa przez niego oferowana czyli Firebase Cloud Storage [9]. Jest to serwis oparty na modelu biznesowym

"pay-as-you-go", czyli opłaty za działanie tego serwisu są proporcjonalne do zużycia zasobów. Dzięki temu, możemy skalować miejsce w chmurze proporcjonalnie do liczby użytkowników i zużywanych przez nich zasobów co spowoduje minimalizację kosztów na początku istnienia systemu. Dodatkowo, Firebase ma możliwość udostępniania adresów URL do zdjęć wraz z kluczem dostępu podpisanym na określony czas. Dodatkowo zdjęcia te nie są indeksowane przez wyszukiwarki tj. Google, dzięki czemu nie można do nich uzyskać dostępu poza naszą aplikacją. Dzięki tym funkcjom, jesteśmy w stanie zaimplementować system przesyłania i udostępniania zdjęć do których będą mieli dostęp jedynie określeni użytkownicy. Na decyzję w wyborze serwisu Firebase miała również wpływ łatwość implementacji i integracji tego serwisu z naszą aplikacją oraz gwarancja niezawodności wynikająca z faktu, że serwisem tym zarządza Google, czyli obecnie jedna z największych firm technologicznych.

### 2.2.5 Frontend

Jako język programowania, zdecydowaliśmy się na TypeScript, jako że zapewnia bezpieczeństwo typów poprzez statyczne typowanie. Zwiększa to niezawodność aplikacji, zwłaszcza przy bardziej złożonych projektach.

W kontekście narzędzia do budowania aplikacji wybraliśmy Vite [10], jako że zapewnia bardzo szybkie budowanie i odświeżanie aplikacji podczas pracy. Przekłada się to na znaczące przyspieszenie implementacji, szczególnie w porównaniu do starszych alternatyw, jak na przykład Webpack.

React [11] jest najczęściej używaną biblioteką wśród narzędzi służących do budowania aplikacji typu SPA (Single Page Application) oraz ma dużą społeczność, co oznacza dostęp do wielu gotowych rozwiązań. Dodatkowo nie bez znaczenia okazało się doświadczenie, które właśnie w React mieliśmy zdecydowanie największe jako zespół. Dlatego też zdecydowano o wybraniu właśnie tej biblioteki.

Wśród innych bibliotek wykorzystywanych w projekcie, znalazł się Ant Design [12], służący do projektowania interfejsów użytkownika, rozpowszechniany na licencji o otwartym kodzie źródłowym. Ant Design zapewnia wiele gotowych komponentów takich jak: formularze, alerty czy komponenty wspomagające zarządzanie układem strony, które znacząco przyspieszają implementację.

Do zarządzania stanem aplikacji, w przypadku obiektów które są współdzielone przez wiele widoków, użyliśmy biblioteki MobX [13]. Jest to to jedna z najpopularniejszych bibliotek do zarządzania stanem, która operuje na zasadzie reaktywności - każda zmiana stanu aplikacji powoduje błyskawiczną aktualizację w komponentach zależnych od tego stanu.

### 2.2.6 OpenCage Geocoding API

W naszym projekcie chieliśmy rozszerzyć tradycyjną formę prezentacji informacji o lokalizacji w formie tekstowej o formę interaktywnej mapy. Zdecydowaliśmy się na integrację z Opencage Geocoding API [14] ze względu na jego funkcjonalność oraz elastyczność kosztową. Darmowy plan zezwala na dwa tysiące pięćset zapytań dziennie, co w pełni spełnia nasze potrzeby na etapie prototypowania i wstępnego rozwijania aplikacji. Dodatkowo, w przyszłości, gdy nasze potrzeby wzrosną, istnieje możliwość rozszerzenia planu na wersję płatną, co pozwoli na większą liczbę zapytań i dodatkowe funkcjonalności, dostosowane do rozwoju aplikacji.

OpenCage Geocoding API umożliwiło nam przechowywanie przybliżonych lokalizacji usługodawców oraz prezentację ich na mapie, co zwiększa komfort i intuicyjność obsługi aplikacji przez użytkowników. Dzięki temu będą oni mogli łatwiej wyszukiwać usługi w wybranej lokalizacji oraz szybciej podejmować decyzje, co pozytywnie wpłynie na ich ogólne doświadczenie i satysfakcję z korzystania z aplikacji.

### 2.2.7 Docker i DockerHub

W celu zapewnienia niezależności działania aplikacji od systemu i ułatwienia wdrażania aplikacji zdecydowaliśmy się na konteneryzację implementowanych przez nas serwisów. Zadanie to realizuje dla nas Docker, który jest narzędziem pozwalającym na łatwe wdrażanie aplikacji w izolowanych środowiskach zwanych kontenerami. Utworzone obrazy docelowo będą przechowywane na platformie DockerHub, dzięki temu będą dostępne online w celach zarówno hostowania jak i uruchamiania na lokalnej maszynie. Dodatkowo utworzone kontenery wspierają zmienne środowiskowe, co zwiększa bezpieczeństwo z uwagi na możliwość przechowywania zmiennych wrażliwych aplikacji tj. hasła i klucze API w osobnych plikach lub bezpośrednio w zmiennych środowiskowych platformy na której wdrażane będą skonteneryzowane aplikacje.

### 2.2.8 Github

W projekcie korzystamy również z narzędzi wspierających efektywną współpracę zespołową nad kodem. Głównym rozwiązaniem w tym zakresie jest GitHub, który pełni funkcję platformy do kontroli wersji,

zarządzania kodem źródłowym oraz ułatwia koordynację prac zespołu. Posiada on również narzędzia służące do ciągłej integracji i ciągłego wdrażania, które używane są do budowania, testowania i tworzenia kontenerów aplikacji ilekroć pojawia się nowa wersja tworzonego przez nas oprogramowania.

## 2.3. Ograniczenia czasowe, zasoby, problemy

Początkowo, mieliśmy w planach wdrożenie integracji płatności. Chcieliśmy umożliwić właścicielom dokonywanie płatności za opiekę nad zwierzęciem bezpośrednio w aplikacji. Pomysł musieliśmy szybko porzucić ze względu na problemy, które napotkaliśmy:

1. **Problemy techniczne i prawne** - w większości przypadków, dostawcy usług płatniczych wymagają podania NIP-u (Numeru Identyfikacji Podatkowej) przy integracji z ich systemem. Jest to projekt inżynierski, a ponadto żaden z autorów projektu nie posiada i nie chciałby zakładać własnej działalności gospodarczej jedynie w tym celu. Wymagałoby to także spełnienie szeregu wymogów bezpieczeństwa. W związku z tym, już na tym etapie wykluczyło to możliwość wdrożenia integracji płatności w naszej aplikacji.

2. **Ograniczenia czasowe** - ze względu na ograniczenia czasowe, wynikające z ustalonego harmonogramu, nie bylibyśmy w stanie zrealizować integracji systemu płatności. Pełne przetestowanie tej funkcji wymagałoby dodatkowych tygodni pracy, co wykracza poza ramy czasowe, które mamy dostępne w ramach projektu.

3. **Zasoby i priorytety** - głównym celem projektu jest umożliwienie efektywnej komunikacji pomiędzy właścicielami zwierząt, a potencjalnymi opiekunami. Wdrożenie płatności, choć ułatwiłoby cały proces, nie jest w naszym przypadku pożądane.

W związku z powyższymi problemami uznaliśmy, że integracja płatności jest elementem, który mógłby zostać zaimplementowany dopiero w przyszłych wersjach aplikacji. W szczególności gdyby zdecydowano, aby wdrożyć nasz produkt komercyjnie.

# 3. Wyniki

## 3.1. Zaimplementowane funkcje

1. **Wyszukiwanie opiekunów** - narzędzie do sprawnego wyszukiwania opiekunów według zadanych kryterów. W ramach wyszukiwania, użytkownik może przeglądać opiekunów na liście, a także wyświetlić ich przybliżone lokalizacje na mapie. Możliwe jest sortowanie opiekunów po średniej ocen, liczbie ocen oraz wskaźniku oceny, wyliczanym na podstawie liczby ocen i średniej. Dostępne jest filtrowanie po następujących polach:

   · dane personalne opiekuna - imię i nazwisko,
   · lokalizacja opiekuna - miejscowość i województwo,
   · cena opieki,
   · rodzaje zwierząt i ich cechy np. wielkość, wiek,
   · udogodnienia np. zabezpieczony ogród, akwarium.

2. **Czat** – system do komunikacji między użytkownikami w czasie rzeczywistym. Możliwa jest konwersacja pomiędzy klientem, a opiekunem także przed złożeniem prosby o opiekę. Obsługiwany jest także przypadek, kiedy użytkownicy znajdują się w różnych strefach czasowych. W oknie chatu znajduje się również opcja blokowania użytkownika.

3. **Profil opiekuna** – funkcja założenia profilu opiekuna oraz zarządzania nim. W tym także możliwość dodawania zdjęć, które wyświetlają się na profilu opiekuna.

4. **Oferta** – zarządzanie ofertą przez opiekuna. Oferta może obejmować udogodnienia i jest możliwa do zrealizowania w podanych przez opiekuna terminach. W ramach jednej oferty opiekun może zdefiniować wiele konfiguracji- zbiorów cech specyficznych dla danego typu zwierzęcia wraz z ceną za jeden dzień usługi.

5. **Opieka** – zarządzanie opieką i jej rezerwacją zarówno przez klienta jak i opiekuna. Możliwość negocjacji ceny oraz obsłużenie różnych scenariuszy, takich jak anulowanie opieki czy jej przedawnienie. Prezentacja historii opieki na linii czasu. Dodatkowo zapewnienie możliwości przeglądania i filtrowania swoich opiek.

6. **Opinie** – umożliwienie wystawiania opinii opiekunom przez klientów, którzy korzystali z ich usług, a więc po zakończonej opiece. Wyraża ocenę jakości świadczonej usługi opieki. Opinia zawiera ocenę liczbową w skali 1-5 oraz opis.

7. **Dodawanie opiekunów do ulubionych** - funkcja umożliwiająca klientom zapamiętanie wybranych opiekunów
i łatwy dostęp do nich, dzięki możliwości ich szybszego wyszukiwania.

8. **Powiadomienia** – wysyłanie powiadomień w przypadku ważnych zdarzeń w aplikacji, np. zmiana statusu opieki.

9. **Bezpieczeństwo aplikacji** – zapewnienie bezpieczeństwa aplikacji, m. in. przez token CSRF oraz integracja z keycloak'iem jako serwisem autoryzującym. Dodatkowo zapewnienie logiki do zarządzania profilami użytkownika.

10. **Blokowanie** – system do blokowania innych użytkowników, aby uniemożliwić im wysyłanie wiadomości oraz wszelkich interakcji związanych z rezerwacją opieki i opiekami, w tym również anulowanie wszystkich rezerwacji opieki przed ich ostatecznym zatwierdzeniem, w stosunku do blokujących.

## 3.2. Spełnione wymaganie niefunkcjonalne

Wymagania niefunkcjonalne dotyczą aspektów technicznych i jakościowych aplikacji. Wykonanie ich jest niezbędne do prawidłowego działania systemu. Obejmuje to między innymi: wydajność, kompatybilność czy dostępność. W ramach naszego projektu zdecydowaliśmy się, aby aplikacja spełniała następujące wymagania:

1. Aplikacja musi działać poprawnie i być kompatybilna z podanymi wersjami przeglądarek internetowych:

   - Google Chrome: wersja 126 i nowsza
   - Mozilla Firefox: wersja 126 i nowsza
   - Microsoft Edge: wersja 126 i nowsza
   - Opera: wersja 112 i nowsza
   - Safari: wersja 16.5 i nowsza

2. Do korzystania z aplikacji wymagane jest stałe łącze internetowe.

3. Aplikacja musi wspierać dwa języki: polski oraz angielski.

4. Interfejs aplikacji musi być prosty i intuicyjny.

5. Aplikacja musi działać poprawnie na urządzeniach mobilnych, tabletach i komputerach. W szczególności obejmuje to responsywność projektowanego interfejsu.

6. Maksymalny czas bezczynności użytkownika wynosi 30 minut, po czym następuje automatyczne wylogowanie.

7. Wdrożenie ochrony przed atakami CSRF.

8. Dane użytkowników powinny być przechowywane w sposób bezpieczny.

9. Dane wrażliwe (np. hasła) muszą być hashowane.

## 3.3. Potencjalne korzyści dla użytkowników

1. Znajdowanie odpowiedniego opiekuna dla zwierzęcia w prosty sposób.

2. Możliwość zaoferowania usługi opieki przez opiekunów.

3. Efektywna komunikacji między stronami opieki poprzez czat.

4. Budowania zaufania do opiekunów na podstawie opinii.

# 4. Wnioski

## 4.1. Podsumowanie wyników

Podsumowując, zaimplementowaliśmy system ułatwiający komunikację między właścicielami zwierząt domowych, a potencjalnymi opiekunami. Zaprezentowane funkcje systemu pokrywają proces zapewniania opieki nad zwierzęciem, od wyszukiwania opiekunów przez właściciela, przez odbywanie opieki, aż po wystawianie opinii. Dodatkowo, system zapewnia dobrą komunikację między właścicielami zwierząt a opiekunami. Wbudowane funkcje, takie jak czat, możliwość blokowania użytkowników oraz dodawania zdjęć ofertowych i profilowych, umożliwiają wygodną i bezpieczną wymianę informacji oraz uwiarygodnienie profili użytkowników. Dzięki tym elementom użytkownicy mogą oczekiwać intuicyjnego i przyjaznego interfejsu, co może pozytywnie wpłynąć na ich doświadczenie z korzystania z aplikacji.

## 4.2. Kierunki rozwoju

Wyszczególniliśmy następujące kierunki rozwoju, które mogłyby być zaimplementowane w przyszłości:

1. Integracja płatności.

2. Statystyki finansowe opiekuna na podstawie płatności w aplikacji.

3. Możliwość zapamiętania przez aplikację swoich zwierząt jako właściciel, aby uprościć proces rezerwacji opieki.

4. Mechanizmy kontroli przed naduzyciami, np. zgłaszanie użytkowników i platforma do moderacji zgłoszeń oraz wykrywanie niedozwolonych zdjęć, obraźliwych opinii itp..

5. Kalendarz opiek opiekuna w celu intuicyjnej organizacji planu.

6. Stworzenie aplikacji mobilnej dzięki której dotrzemy do większej liczby użytkowników.

7. Publikacja i wdrożenie aplikacji do użytku publicznego.

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
