# OpenReview forum: "PetBuddy"
_pwr.edu.pl/Wrocław_University_of_Science_and_Technology/2024/ZPI_Day — Wrocław University of Science and Technology 2024 ZPI Day Submission_

### Official Review · Reviewer_boct · 2024-12-04
**Dobra praca inżynierska wspomagająca networking miłośników zwierząt**

**Confidence:** 5
**Significance Of Results:** 5
**Overall Quality:** 4

**Compliance With Template:**

4: High Quality – The article contains all the required sections, which are well-written and substantively correct, although minor errors or shortcomings may be present. The overall structure is clear and coherent.

**Description Of Results:**

4: High Quality – The results are described in detail and supported by usage examples or evaluations. The description is reliable but may lack full depth of analysis.

**Feedback On Consistency:**

Artykuł napisany popranie i spójnie językowo. Cel pracy jasno zdefiniowany oraz zrealizowany. Rozdział 3.3 powinien posiadać tekst wprowadzający przed listą korzyści.
Brak zdjęć przedstawiających opracowany system.

**Potential For Development:**

Praca posiada potencjał do dalszego rozwoju i próby jej komercjalizacji.

**Project Nature Evaluation:**

Praca spełnia w całości wymogi stawiane pracom inżynierskim. Przedstawione rozwiązanie jest nowatorskie. Technologie dobrze dobrane do realizowanego zadania.

**Technical Language Precision:**

5: Very High Quality – The language is entirely appropriate for a technical report. All terms are used correctly and precisely, and the style is professional, clear, and coherent, without any errors or ambiguities.

---

### Official Review · Reviewer_MZFg · 2024-12-06
**PetBuddy Review**

**Confidence:** 5
**Significance Of Results:** 5
**Overall Quality:** 5

**Compliance With Template:**

5: Very High Quality – The article contains all the required sections, which are written in a very detailed, clear, and error-free manner. The structure is professional and meets expectations, and the content adheres to the highest substantive and formal standards.

**Description Of Results:**

4: High Quality – The results are described in detail and supported by usage examples or evaluations. The description is reliable but may lack full depth of analysis.

**Feedback On Consistency:**

Tytuł zawiera niefortunny przecinek. W abstrakcie na szczęście już go nie ma. Powraca w podsumowaniu w sekcji 4.1.

Przyczyna realizacji projektu jest poprawnie uzasadniona. Cel we wstępie jest szerzej wypunktowany. Strategiczne byłoby pozostawienie bardziej ogólnego celu, a uszczegółowienie go dopiero w kontekście wad i zalet rozwiązań konkurencyjnych (przedstawionych w sekcji 2.1).

Jednocześnie opis celu tekstem ciągłym daje poczucie nieuporządkowania. Brakuje prostego diagramu/obrazka/infografiki (lub nawet wypunktowania) mówiącego, w jakich czynnościach planowane rozwiązanie wesprze podstawowe typy użytkowników systemu (ew. jak wesprze różne interakcje miedzy tymi użytkownikami). Chcę zaznaczyć, że wiem że zostaje to później zrealizowane po części w sekcji 3 (Wyniki), ale nadal twierdzę, że byłoby to z korzyścią dla opisu, gdyby było to w uporządkowany sposób zarysowane już na etapie opisującym cel szczegółowy.

Cytowania i sposób ich przywoływania w tekście wyglądają poprawnie.

**Potential For Development:**

Opis proponowanego (i częściowo zrealizowanego) rozwiązania rokuje pozytywnie.
Opis przyszłych funkcjonalności brzmi adekwatnie do potrzeb potencjalnych użytkowników systemu.
Przy rzeczywistym wdrożeniu konieczne jest uwzględnienie uwarunkowań prawnych.

**Project Nature Evaluation:**

Struktura technologiczna projektu wygląda poprawnie i została uzasadniona w opisie.
Zrealizowany zakres funkcjonalny wygląda wystarczająco.

**Technical Language Precision:**

5: Very High Quality – The language is entirely appropriate for a technical report. All terms are used correctly and precisely, and the style is professional, clear, and coherent, without any errors or ambiguities.

---

### Official Review · Reviewer_iCYG · 2024-12-06
**Typowa appka do rezerwacji**

**Confidence:** 5
**Significance Of Results:** 3
**Overall Quality:** 4

**Compliance With Template:**

4: High Quality – The article contains all the required sections, which are well-written and substantively correct, although minor errors or shortcomings may be present. The overall structure is clear and coherent.

**Description Of Results:**

4: High Quality – The results are described in detail and supported by usage examples or evaluations. The description is reliable but may lack full depth of analysis.

**Feedback On Consistency:**

Praca jest spójna i logiczna

**Potential For Development:**

Produktów takich istnieje wiele, bo udane wdrożenie wymaga przyciągnięcia (rejestracji) krytycznej masy użytkowników z obu stron: właścicieli i potencjalnych opiekunów.

**Project Nature Evaluation:**

Jako opis projektu software'owego, dlaczego nie ma tu żadnego diagramu UML, mockupu interfejsu czy choćby zrzutu ekranu?
To by uwiarygodniło, że produkt powstał. A tak jest tylko ładnie opisany pomysł i wymagania.

**Technical Language Precision:**

5: Very High Quality – The language is entirely appropriate for a technical report. All terms are used correctly and precisely, and the style is professional, clear, and coherent, without any errors or ambiguities.

---

### Official Review · Reviewer_GcTY · 2024-12-08
**PetBuddy**

**Confidence:** 4
**Significance Of Results:** 4
**Overall Quality:** 4

**Compliance With Template:**

5: Very High Quality – The article contains all the required sections, which are written in a very detailed, clear, and error-free manner. The structure is professional and meets expectations, and the content adheres to the highest substantive and formal standards.

**Description Of Results:**

4: High Quality – The results are described in detail and supported by usage examples or evaluations. The description is reliable but may lack full depth of analysis.

**Feedback On Consistency:**

Projekt PetBuddy wykazuje dużą spójność w strukturze i treści, co ułatwia zrozumienie jego założeń i funkcji. Wszystkie sekcje są logicznie powiązane, a używana terminologia pozostaje konsekwentna w całym dokumencie. Praca jest wolna od istotnych błędów merytorycznych czy językowych. Bardzo dobrze dobrana literatura oraz opisane potencjalne dalsze kroki rozwoju projektu.

**Potential For Development:**

W pracy błędnie wpisano numer (dla języka polskiego) 18,336,600. Szkoda, że zabrakło w pracy wyników przeprowadzonych badań. Projekt bardzo ciekawy tematycznie.

**Project Nature Evaluation:**

Projekt PetBuddy to aplikacja webowa ułatwiająca organizację opieki nad zwierzętami domowymi poprzez skuteczną komunikację między właścicielami a opiekunami. System umożliwia wyszukiwanie usług opieki według lokalizacji, ocen i preferencji, a także oferuje funkcje takie jak czat, negocjacje cen i zarządzanie ofertami. Dzięki integracji z Keycloak zapewnia bezpieczne logowanie, a zastosowanie Firebase i OpenCage Geocoding API umożliwia przechowywanie danych oraz prezentację lokalizacji na mapie.
Projekt oparty jest na technologiach, takich jak Spring, PostgreSQL i React, zapewnia intuicyjność, wydajność oraz kompatybilność z różnymi urządzeniami. Zgodnie z treścią projektu - PetBuddy wypełnia lukę na rynku usług opieki nad zwierzętami, oferując praktyczne i skalowalne rozwiązanie. Przyszły rozwój, w tym integracja płatności i aplikacja mobilna, dodatkowo zwiększy potencjał komercyjny systemu.

**Technical Language Precision:**

4: High Quality – The language is appropriate for a technical report. Terminology is used correctly, and statements are precise, with only minor shortcomings that do not affect the overall clarity.

---

### Decision · Program_Chairs · 2024-12-10

Accept (Poster)